# Severe Muscle Deconditioning Triggers Early Extracellular Matrix Remodeling and Resident Stem Cell Differentiation into Adipocytes in Healthy Men

**DOI:** 10.3390/ijms23105489

**Published:** 2022-05-14

**Authors:** Corentin Guilhot, Théo Fovet, Pierre Delobel, Manon Dargegen, Bernard J. Jasmin, Thomas Brioche, Angèle Chopard, Guillaume Py

**Affiliations:** 1DMEM, Montpellier University, Institut National de la Recherche pour l’Agriculture, l’Alimentation et l’Environnement (INRAE), 2 Place Pierre Viala, Bat. 22, 34060 Montpellier, France; theo.fovet@umontpellier.fr (T.F.); pierre.delobel@inrae.fr (P.D.); manon.dargegen@orange.fr (M.D.); thomas.brioche@umontpellier.fr (T.B.); angele.chopard@umontpellier.fr (A.C.); 2Department of Cellular and Molecular Medicine, Eric J. Poulin Centre for Neuromuscular Disease, Faculty of Medicine, University of Ottawa, Ottawa, ON K1H 8M5, Canada; jasmin@uottawa.ca

**Keywords:** extracellular matrix, fibro-adipogenic progenitors, intermuscular adipose tissue, microenvironment, muscle wasting, severe muscle disuse, spaceflight

## Abstract

Besides the loss of muscle mass and strength, increased intermuscular adipose tissue (IMAT) is now a well-recognized consequence of muscle deconditioning as experienced in prolonged microgravity. IMAT content may alter the muscle stem cell microenvironment. We hypothesized that extracellular matrix structure alterations and microenvironment remodeling induced by fast and severe muscle disuse could modulate fibro-adipogenic progenitor fate and behavior. We used the dry immersion (DI) model that rapidly leads to severe muscle deconditioning due to drastic hypoactivity. We randomly assigned healthy volunteers (n = 18 men) to the control group (only DI, n = 9; age = 33.8 ± 4) or to the DI + thigh cuff group (n = 9; age = 33.4 ± 7). Participants remained immersed in the supine position in a thermo-neutral water bath for 5 days. We collected vastus lateralis biopsies before (baseline) and after DI. 5 days of DI are sufficient to reduce muscle mass significantly, as indicated by the decreased myofiber cross-sectional area in vastus lateralis samples (−18% vs. baseline, *p* < 0.05). Early and late adipogenic differentiation transcription factors protein levels were upregulated. Platelet-derived growth Factors alpha (PDGFR⍺) protein level and PDGFR⍺-positive cells were increased after 5 days of DI. Extracellular matrix structure was prone to remodeling with an altered ECM composition with 4 major collagens, fibronectin, and Connective Tissue Growth Factor mRNA decreases (*p* < 0.001 vs. baseline). Wearing thigh cuffs did not have any preventive effect on the measured variable. Our results show that altered extracellular matrix structure and signaling pathways occur early during DI, a severe muscle wasting model, favoring fibro-adipogenic progenitor differentiation into adipocytes.

## 1. Introduction

Astronaut health preservation lies in the capacity to understand biological mechanisms involved during spaceflight, especially for planned future long-term space travel beyond the classical 6-month ISS sojourn. Spaceflight is characterized by a microgravity-induced decrease in skeletal muscle solicitation, inducing muscle deconditioning, despite the two h daily physical exercise [1]. Muscle deconditioning is characterized by reduced muscle strength and power, essentially due to muscle mass loss [2]. Muscle deconditioning is a significant healthcare challenge and socioeconomic burden [3], given that it is observed in many contexts, such as aging, diseases, chronic inactivity, and spaceflight.

In the last decade, researchers started to consider muscle as a secretory organ that can interact with other organs and other cell types. Resting and contracting myocytes synthesize and release myokines to interact with other cell types present in muscle, including stem cells, leading to complex and well-orchestrated crosstalk [4,5]. Previous studies have shown the importance of this crosstalk, particularly for proper muscle development [6]. The muscle fiber microenvironment is essentially composed of an extracellular matrix (ECM), defined as a complex network of collagen, hyaluronan, fibronectin, proteoglycans, and scaffold proteins [7], and also composed of many resident stem cells of endothelial or mesenchymal origin [8,9]. ECM structural modifications (i.e., degradation or thickening) could drive functional events that involve stem cell proliferation and differentiation [10]. Therefore, ECM is now considered a plastic and responsive component that acts as a reservoir for several proteins and factors implicated in controlling the growth and differentiation of resident cells. Today, the impact of ECM structure and function on the muscle microenvironment remains understudied.

Among the stem cells present in the muscle microenvironment, PAX7-positive (PAX7^+^) satellite cells (SC) are located at the periphery of muscle fibers under the basal lamina and are required for muscle maintenance [11,12]. Upon injury, quiescent SCs start to proliferate and form new myofibers for muscle regeneration. Increasing evidence shows that during aging, SC proliferation/self-renewal is impaired [13]. Moreover, the SC pool is decreased after 14 days of bed rest [14].

Fibro-adipogenic progenitors (FAPs) constitute a multipotent stem cell population present in the muscle microenvironment [15,16]. These cells express platelet-derived growth factor receptor alpha (PDGFR⍺) at the surface and are known to promote SC differentiation into muscle cells during muscle regeneration [17]. FAPs are of mesenchymal origin and can differentiate at least into the fibrotic and adipogenic lineages [18]. FAPs fate and behavior can be influenced by several signals from the microenvironment. Previous studies suggested a direct interaction between myocytes and FAPs [17]. In vitro myocytes can inhibit FAPs differentiation into adipocytes through myokine signaling [19]. FAPs might promote SC self-renewal through WISP1 [20] and IL−6 and SC differentiation into muscle cells through follistatin in a muscle regeneration model [21]. However, secreted CCN member CCN1 protein induces FAPs differentiation into adipocytes ([22], p. 1). Interestingly, TGFβ and myostatin also promote FAPs differentiation into fibroblast [23,24]. Intermuscular adipose tissue (IMAT) accumulation is a hallmark of deconditioned muscle because it has been observed in several muscle disuse models, such as bed rest and hind limb suspension [25]. Conversely, IMAT content is lower in elite athletes than in their sedentary counterparts [26]. These studies suggest that the level and chronicity of muscle contractile activity are major IMAT regulators.

Cumulative data now shows that aging and chronic degenerative skeletal muscle diseases affect the muscle microenvironment composition, particularly muscle stem cells [20,27]. However, whether microgravity-induced hypoactivity can alter the muscle microenvironment is not known. Extrinsic and intrinsic factors can influence the functions of muscle cell progenitors (SCs) [28]. Among the extrinsic factors, ECM structural and signaling changes play a critical role in SC fate and behavior [29,30]. Whether similar mechanisms could regulate FAPs fate and behavior has to be clarified. In this study, we used a ground-based model of microgravity, the dry immersion (DI) model that rapidly induces severe muscle deconditioning, to question whether ECM remodeling exists after only five days of severe inactivity. We further hypothesized that early changes in muscle microenvironment structure and signaling might influence SCs and FAPs stem cell fate and behavior.

## 2. Results

Of the twenty selected subjects, two were excluded for reasons unrelated to the protocol used in the study. Thus, a total of 18 healthy men were included in the present study and were randomly divided into the Control and Cuffs groups (n = 9/group). Since our previous work on the same subject did not show any significant difference in atrophy and muscle strength loss between CTL and Cuffs groups [31], we arbitrarily chose to pool the data of all participants (n = 18) to increase the statistical power. However, due to limited biopsy materials, the number of subjects for each analysis is indicated in the figure legend.

### 2.1. Dry Immersion for 5 Days Leads to ECM Remodeling

The ECM plays a key role in controlling the skeletal muscle microenvironment because it can adapt to different constraints and influence the resident stem cell crosstalk [20,22,27]. In general, the ECM and its impact on muscle remodeling remain understudied.

In the current study, we first assessed ECM structural changes were assessed by Sirius red staining. Despite a tendency to decrease in total collagen surface area using Sirius Red staining, we retrieved significant decreases in collagen 1a protein and mRNA. Interestingly, the basal level of fibrosis was significantly decreased after 5 days of DI (1.5% vs. 2.5% Post vs. Pre DI *p* < 0.05) (Figure 1a,b).

In agreement with this finding, the expression of mRNAs encoding fibronectin and ⍺SMA was significantly decreased after 5 days of DI (−38% ± 23% and −56% ± 15%, respectively, *p* < 0.05 vs. Pre-DI for both). We also focused on the main collagen form in ECM, and we showed a not significant decrease in mRNA (−23% ± 45%, *p* = 0.1) (Figure 1c) but a significant decrease in Collagen 1a protein levels after 5 days of DI (−22% ± 14%, *p* = 0.01) (Figure 1b). Expression of mRNAs encoding other major collagen types in the ECM was also decreased at DI5 compared with Pre-DI: collagen IIIa (−52% ± 45%, *p* < 0.05), collagen VIa2 (−24% ± 34%, *p* < 0.05), and also collagen IV (−21% ± 25% *p* < 0.05), the most abundant collagen type in basal lamina (Figure 1c). These results indicate that after 5 days of DI, the ECM starts to become disorganized (reduced synthesis of reticulated collagen), although no macroscopic change could be detected.

### 2.2. ECM Signaling Alterations

ECM structural changes can influence signaling pathways and growth factor bioavailability. After 5 days of DI, the expression of mRNAs encoding ECM modulators, such as Connective Tissue Growth Factor (*CTGF/CCN2*; −38% ± 27% *p* < 0.05), expressed in myofibroblast [33], and Tissues Inhibitor of Metalloprotease 3 (*TIMP3*; −13% ± 29% *p* = 0.09), was decreased compared with the Pre-DI muscle samples (Figure 2a). As ECM disorganization may affect the bioavailability of other growth factors that influence vascular remodeling and stem cell fate and behavior, mRNA expression of Vascular Endothelial Growth Factor (*VEGF*) (−23% ± 29% vs. Pre-DI, *p* < 0.05) and Insulin Growth factor 1 (*IGF1*) (−22%, ± 21% vs. Pre-DI, *p* < 0.05) was downregulated (Figure 2b). Inhibitor of FGF signaling Sprouty1 protein level also was reduced at DI5 (−33% ± 34%, *p* = 0.09 vs. Pre-DI) (Figure 2c). Conversely, mRNA expression of Fibroblast Growth Factor 2 (*FGF2*; +11% ± 35%, *p* < 0.05 vs. Pre-DI) and myostatin (*MSTN*; +37% ± 69%, *p* < 0.05) was increased. These results support the notions that FGF2 signaling could reduce SC quiescence, possibly through inhibition of Sprouty 1 signaling [34], and that an increased level of Mstn may induce muscle myocyte atrophy and reduce SC fate and behavior [35]. Taken together, these data suggest a shift in the muscle microenvironment towards signals that may promote muscle atrophy, devascularization, and induction of FAPs differentiation.

### 2.3. PAX7^+^ Cells Population Is Decreased after 5 Days of DI

With myopathies or aging, SCs lose their stemness and fuse with the adjacent myofibers [36] without proliferating, leading to a decreased regenerative potential [37]. PAX7 staining showed a significant decrease in staining intensity after 5 days of DI (−29%, ± 17% *p* < 0.05 vs. Pre-DI). In agreement with this, Western blot analysis also revealed a reduction in PAX7 protein levels (Figure 3a–d). Conversely, *PAX7* mRNA expression was not significantly downregulated after 5 days of DI (−19.5% ± 32%, *p* = 0.1, Figure 3d). Finally, protein levels of the transcription factors MYOD and myogenin, both involved in SC proliferation and myogenesis [38,39], were not affected by 5 days of DI (Figure 3c).

### 2.4. PDGFRα Expression Is Increased after 5 Days of DI

FAPs are considered the main source of muscle IMAT and are identified via PDGFRα expression. At DI5, PDGFRα protein expression was significantly increased in muscle samples as measured by western blotting (+45% ± 110% *p* < 0.05 vs. Pre-DI) (Figure 4a) and immunohistochemistry (+69% ± 210% *p* < 0.05 vs. Pre-DI) (Figure 4b,c). FAPs may dynamically generate cilia and express the Intra Flagellar Transport 88 (*IFT88*) surface protein. *IFT88* mRNA expression was slightly but significantly increased (+9% ± 18% < 0.05 vs. Pre-DI) at DI5 (Figure 4c). These results clearly show an increase in the number of PDGFR⍺^+^ FAPs and suggest their commitment to adipogenic differentiation.

### 2.5. Fibro-Adipogenic Progenitor Fate after 5 Days of DI

To confirm FAPs commitment to adipogenic differentiation, the expression of different adipogenic markers was investigated. Protein expression of C/EBPβ, which plays an important role in the early stage of adipocyte progenitor differentiation, was significantly increased at DI5 (+31% ± 53%; *p* < 0.05 vs. Pre-DI) as well as C/EBP⍺, major late marker of adipogenesis (+37% ± 54%, *p* < 0.05 vs. Pre-DI), PPAR𝛾 was also upregulated (+29%, *p* = 0.09) (Figure 5a) [40]. This is consistent with the proliferation and differentiation into adipogenic cells of a stem cells population that includes at least FAPs. To determine whether fully differentiated adipocytes were increased at DI5, the expression of fatty acid-binding protein 4 (FABP4) and perilipin was assessed by western blotting in Pre-DI and Post-DI muscle samples. Both proteins were increased by +34% ± 116% and +29% ± 74%, respectively, in Post-DI compared with Pre-DI samples (*p* < 0.05) (Figure 5b). Altogether, these results suggest that a short period of drastic inactivity initiates cellular and molecular events such that FAPs progenitors proliferate and differentiate at least in part into the adipogenic lineage. However, due to the short-lasting period, these cellular events do not translate into macroscopic ectopic fat content when measured by MNR (Figure 5c).

## 3. Discussion

Over the past two decades, researchers have described IMAT appearance as a hallmark in many situations of muscle deconditioning, including aging, chronic degenerative muscle disease, and muscle deconditioning models. Although IMAT appearance can be influenced by pathophysiological features, fat infiltration seems to be strongly linked to sedentary and is now considered an indicator of poor muscle function [41]. Since extrinsic and intrinsic factors can influence the functions of muscle cell progenitors (SCs) [28], we hypothesized that MD-induced ECM remodeling could also influence FAPs fate and behavior leading to ectopic fat deposits. We found that 5 days of DI are sufficient to modify ECM structure and signaling that are associated with FAPs commitment toward adipogenesis (Figure 6).

Long-duration ground-based microgravity models induce muscle mass loss, muscle fiber atrophy, and muscle strength decrease. Conversely, few data are available on the effects of short inactivity periods. Edgerton et al. [42] showed a mean CSA decrease of 11% and 24% for type I and II muscle fibers, respectively, after 5 days of space flight. Shenkman et al. [43], using the DI model, observed atrophy of both type I and II muscle fibers (5–9% of CSA decrease after 3 days and 15–18% after 7 days of DI). More recently, Pagano et al. [44] reported a significant decrease in VL myofiber CSA (−10.6%) after 3 days of DI. In a previous work of the same experiment, we found a decrease of 18% in VL myofiber CSA after 5 days of DI as well as a mean 12% decrease in maximal knee extension isometric torque (Appendix A). These findings confirm that this model consistently and reproducibly induces rapid atrophy [31,44,45].

The origin of muscle atrophy is multifactorial and implicates the downregulation of anabolic signals and the upregulation of catabolic signals. We first show here that 5 days of DI are sufficient to decrease *IGF1* expression significantly. IGF1 paracrine and autocrine actions regulate muscle fiber size, but its production and secretion are impaired in conditions of muscle disuse [46]. Conversely, *MSTN*, a key negative regulator of muscle mass with a major muscle atrophic effect, was upregulated, as previously reported by Wall et al. [47], who found that *MSTN* gene expression doubled after 5 days of one-legged knee immobilization. This was associated with reduced muscle fiber CSA and a 12% loss of VL extensor isometric strength. Among studies that determined both muscle mass and strength, some reported that strength loss was more important than muscle mass reduction. Conversely, we found a similar loss of muscle mass (−18% CSA) and strength (−14%) as in our previous study on the effects of 3 days of DI (−10% of CSA and −9% of VL strength) [48].

Recent literature data indicate that IMAT development is a key feature of muscle deconditioning and is strongly linked to muscle atrophy. We hypothesized that muscle microenvironment reorganization during an intense period of inactivity, as experienced during DI, favors IMAT appearance. The ECM is a plastic and responsive tissue sensitive to stress intensity and duration [49]. For instance, a long inactivity period and aging lead to ECM thickening, altered collagen deposits, and fibrosis [50]. These alterations are correlated with loss of function [10]. We did not observe any significant change in *COL1A1* gene expression or Sirius red-stained area after 5 days of DI. Conversely, 5 days of DI were sufficient to decrease basal fibrosis [51] and ECM composition since the expression of CTGF/CCN2, collagens, fibronectin, and αSMA genes were all downregulated. These results are consistent with those of our previous study after 3 days of DI. It suggests that ECM remodeling can occur rapidly in the first few days when using a drastic muscle wasting model like DI.

ECM is a reservoir of several factors that influence the muscle environment and a key regulator of the fate of muscle resident stem cells [10]. ECM density and composition influence resident stem cell fate, growth factor availability, and local signaling [10,20,27,52]. VEGF, a pro-angiogenic growth factor, is produced by contracting myocytes and is implicated in SC niche maintenance by preventing their early entry into the cell cycle [53]. Interestingly, VEGF secretion in muscle is impaired by aging and inactivity [54]. In agreement, we found that *VEGF* gene expression was decreased at DI5. Conversely, *FGF2* and *MSTN* were upregulated at DI5, and the protein level of Sprouty 1, which is needed to maintain SC quiescence, was decreased [55]. This may lead to an imbalance between beneficial and harmful microenvironment factors. Interestingly, FGF2 is constitutively upregulated in the muscle microenvironment of aged mice. This promotes spontaneous SC activation and fusion with adjacent myocytes, but without proliferation, leading to SC pool depletion and contributing to impairing the resident stem cell crosstalk [56]. PAX7⁺ muscle progenitors (SCs) also were reduced at DI5. Their capacity to multiply asymmetrically for self-renewal is crucial for muscle homeostasis. However, in muscle-wasting conditions, SCs lose this capacity, and this might contribute to the decreased regeneration capacity. Guitart et al. [57]. showed that 7 days of immobilization in rats induced loss of PAX7⁺ cells in gastrocnemius and soleus. Arentson-Lantz et al. [14] also found that SCs were decreased by −39% after 14 days of bed rest. Therefore, alterations in the balance between healthy and atrophic signals and the decrease in PAX7 signal suggest an early effect on the signaling regulating SC fate and behavior that contributes to their decrease.

Among the atrophic signals with a direct effect on stem cells, myostatin influences SCs and also FAPs [23]. Specifically, in chronic kidney disease, myostatin can drive FAPs towards the fibrotic lineage [23]. Although *MSTN* was clearly upregulated in the DI model, its role in FAPs engagement towards the fibrotic lineage is less clear than in chronic kidney disease because we also observed a strong upregulation of adipogenic transcription factors. The choice of FAPs fate is influenced by many intrinsic and extrinsic factors that can be modulated by muscle microenvironment changes [58]. We cannot exclude that a longer DI duration, with continuous *MSTN* upregulation, might drive FAPs differentiation into fibroblast. Indeed, after 4 weeks of cast immobilization, rat muscle atrophy was accompanied by increased αSMA, collagen I and III, and TGF-β1 content [59]. As we retrieved a decreased aSMA, CTGF, main collagen mRNA isoforms along with decreased Col1A protein content and reduced fibrosis area, it seems that microenvirenment changes preferentially drive FAPs into adipogenic lineage instead of myofibroblast one, at least in the early stage of deconditioning. Interestingly, aging muscle is accompanied by increased fibrosis and IMAT content, but whether a prolonged period of severe inactivity in healthy adults can preferentially orient FAPs towards myofibroblast lineage is not clear. Thus, even if some microenvironment signaling pathways (FGF2 and Mstn) favor FAPs orientation towards myofibroblast lineage, it seems that ECM signaling remodeling is in favor of adipogenic FAPs orientation in the present model.

Similar to our previous study (3 days of DI) [44], 5 days of DI were sufficient to observe an increase in PDGFR⍺⁺ cells and the expression of most of the early and late markers of adipocyte differentiation (C/EBPβ, PPAR𝛾, and C/EBP⍺). The increase in FABP4 and perilipin 1 protein levels in muscle biopsies at DI5, two markers of mature adipocytes (+34% and +29%, respectively, vs. Pre-DI), suggests that some FAPs could commit directly to the adipogenic lineage without proliferating first. Manini et al. [60] showed that 4 weeks of lower limb suspension impairs muscle volume and leads to IMAT appearance in the thigh (14.5%) and calf (20%) muscles of young adults (MRI measurement).

FAPs might be sensitive to ECM modulation, but few data are available. Specifically, their differentiation seems to be induced by MMP14 [27] and inhibited by TIMP3 [58]. Kopinke et al. [58] demonstrated that FAPs could produce primary cilia in some conditions. They showed that ciliation inhibition, through the downregulation of *IFT88* (required for ciliogenesis), restrains FAPs differentiation towards adipocytes. Here, we observed *IFT88* upregulation and *TIMP3* gene downregulation (not significant) in Post-DI compared with Pre-DI samples; one can argue that upregulation of IFT88 could be linked to increased ciliogenesis and FAPs engagement towards adipogenesis after 5 days of DI. We conclude that significant changes in an extracellular matrix structure and signaling pathways occur after only 5 days of severe muscle inactivity (Dry immersion model) and that these changes can modulate satellite cells and fibro-adipogenic progenitors’ fate and behavior, leading to a decrease in the satellite cells pool and an increase in adipogenesis. Future studies should determine the cellular origin and the precise temporal kinetics of the different signals implicated in the complex crosstalk of myocytes, satellite cells, and fibro-adipogenic progenitors.

## 4. Material and Methods

### 4.1. Subjects and Ethics Considerations

All experiments were performed at the Space Clinic of the Institute of Space Medicine and Physiology (Medes-IMPS, Rangueil Hospital) in Toulouse (France) and were sponsored by the French National Space Agency (CNES). This study was carried out from 19 November 2018 to 23 March 2019. Twenty healthy men were selected for this experiment (age: 32 ± 5 years; height: 179 ± 7 cm; weight: 74.5 ± 7.2 kg; BMI: 23.5 ± 1.6). Participants did not have any history or physical signs of neuromuscular disorders, were non-smokers, and did not take any drug or medication. They all gave their written informed consent to the experimental procedures that were approved by the local ethics committee (CPP Est III: 2 October 2018, n° ID RCB 2018-A01470-55) and French Health Authorities (ANSM: 13 August 2018) and were in accordance with the Declaration of Helsinki. This study was registered at ClinicalTrials.gov (Identifier: NCT03915457).

### 4.2. Study Design, Dry Immersion Protocol, Thigh Cuffs

The experimental protocol included ambulatory baseline measurements, followed by 5 days (120 hours) of DI and two days of ambulatory recovery. A week before the protocol initiation, participants went to Medes to undergo a Pre-DI muscle biopsy and baseline measurements (supplementary data from Fovet et al. [31]). Participants were randomly assigned to the control group (CTL) or the thigh cuff group (Cuffs). Participants randomized in the Cuffs group wore thigh cuffs on both legs during the 5 days of DI: from 10 a.m. to 6 p.m. at day 1 of DI (DI1) and from 8 a.m. to 6 p.m. at DI2-DI5. Thigh cuffs were adapted to each man by performing calf plethysmography in the supine position before immersion to obtain a lower-limb counterpressure of about 30–50 mmHg. At DI1, thigh cuffs were put on immediately before immersion. DI was performed according to the methodology detailed by [61]. Two participants (n = 1 Control and n = 1 Cuffs) underwent DI simultaneously in the same room, in two separate baths. The water temperature was continuously maintained thermoneutral. The light-off period was from 11:00 p.m. to 7:00 a.m. Daily hygiene, weighing, and some specific measurements required extraction from the bath. During these out-of-bath periods, participants maintained the −6° head-down position. The total out-of-bath supine time during DI was 1.1 ± 0.6 h for DI1–4 and 5.3 ± 1.1 h for DI5 because of muscle biopsy and MRI. Otherwise, during DI, participants remained immersed in the supine position for all activities and were continuously observed by video monitoring. Bodyweight, blood pressure, heart rate, and tympanic body temperature were measured daily. The adequate water intake was fixed at 35–60 mL·kg^−1^day^−1^; within this range, water intake was ad libitum (measured). The menu composition of each DI day was identical for all participants. Dietary intake was individually tailored and controlled during the study.

### 4.3. Muscle Biopsies

Two right vastus lateralis (VL) muscle biopsies were performed. The first was done before DI (Pre-DI, mean weight = 355.4 ±95 mg) and the second at the end of DI5 and near the first puncture site (Post-DI, mean weight = 335 ± 106 mg). Biopsies were performed following a well-established procedure using a 5 mm Bergström biopsy needle in sterile conditions and under local anesthesia (1% lidocaine) [62]. Biopsies were cut into several pieces. One was immediately frozen in liquid nitrogen, and another was embedded in a small silicone cast that was filled with OCT embedding medium (Cat#: Lamb/OCT, ThermoScientific, Waltham, MA, USA) and immediately frozen in liquid nitrogen. These two pieces were stored at −80 °C. A third piece was fixed in 4% paraformaldehyde solution (Cat#: 526936, Sigma-Aldrich, Saint-louis, MI, USA) at room temperature (RT) overnight and then embedded in paraffin.

### 4.4. Magnetic Resonance Imaging (MRI)

Siemens-Avanto device was used in CHU Rangueil in Toulouse, France, to perform images. Subjects were transferred to an MRI room keeping −6° head-down bed rest position to simulate microgravity in the best way. Imaging was recorded 4 days before immersion and directly after. Briefly, the mark was located exactly halfway between the left anterior superior iliac spine and the upper part of the ipsilateral (left) patella.

The measurements were taken on the left side because of the biopsy on the right side. This condition was chosen to avoid inducing heterogenic signals in MRI analysis of the biopsy site. The fat content was calculated by OsiriX MD software (v.7.0.1. 64-bit software, managed by Rosset A., Genève, Switzerland). Three-point Dixon MRI was used to measure intramuscular fat content (IFC) as a percentage inside the anterior muscle compartment from the images. IFC was measured in the same region of interest (ROI) on images centered on the same landmark. The measurements were repeated until agreement was within the 3% threshold, and the values were averaged.

### 4.5. Pax7 Staining

Pre-DI and Post-DI VL serial transverse cryosections (10 μm thick), cut using a cryostat at −25 °C, were dried and fixed in acetone for 10 minutes before washes in phosphate-buffered saline (PBS), blocking, and permeabilization with 0.1% Triton-X100 (Cat#: 11332481001, Sigma-Aldrich, Saint-louis, MI, USA) and 20% horse serum. To evaluate SC stemness, Pax7 staining was performed on serial transverse cryosections. Sections were incubated with an anti-Pax7 (Table 1) at 4 °C overnight and then with the relevant secondary antibody (Table 1) at 37 °C for 1 hour.

Nuclei were stained with Hoechst solution (1/1000) for 30 seconds. Then, sections were mounted with Permafluor^®^ (Cat#: TA-030-FM, Epredia, Thermo fisher scientific, Waltham, MA, USA) and dried at RT overnight. Images were obtained using a ZEISS Axio Scan (INM, Montpellier, France) at ×20 magnification with a focus on the DAPI-stained nuclei. Signal quantification was analyzed using ImageJ (version 1.46).

### 4.6. ECM Remodeling Analysis

Sirius red staining was performed according to Junqueira et al. [63] et al. to quantify ECM collagen area and fibrosis. We developed a macro based on polarized light microscopy [32]. Sirius red staining and polarized light optics allow following the maturation of fine collagen fibers into thicker collagen bundles by detecting changes in the birefringence pattern; exclusion was manually performed. Muscle cryosections were dried for 15 to 20 minutes, then fixed in 4% paraformaldehyde for 10 minutes. After two washes, sections were stained with Sirius red solution (0.1% in picric acid) for 1 hour. After two acid-saturated H₂O washes, sections were dried and then incubated in 100% ethanol and xylene before mounting in Permafluor^®^ and overnight drying at RT. Images were obtained with a Leica THUNDER microscope and analyzed with ImageJ (2.0.0-rc-65 version).

### 4.7. PDGFRα Staining

A piece of each muscle biopsy was fixed in 4% paraformaldehyde (24 h) and paraffin-embedded to prepare 3 µm thick sections. PDGFRα staining was performed as described by Pagano et al. [44] using an HMS740 Robot Stainer. Briefly, sections were deparaffinized, rehydrated, and incubated in EDTA buffer (pH 9) at 95 °C for 10 minutes. Sections were washed in PBS and incubated in 0.3% H_2_O_2_ at RT for 20 minutes. Endogenous biotin was blocked using an Avidin/Biotin Blocking Kit (Cat#: System SP-2001, Vector Laboratories, Burlingame, CA, USA), and non-specific binding was blocked by incubation in TBS containing 20% goat serum at RT for 30 minutes. Sections were incubated at 4 °C with an anti-PDGFRα antibody (1:50 in TBS/5% goat serum) overnight, followed by washes with PBS and incubation with an anti-rabbit IgG secondary antibody (1:500) at RT for 45 minutes. Antibody binding was revealed using the avidin-biotin complex method and the ABC Elite Vectastain Kit (Cat#: PK-6101, Vector Laboratories, Burlingame, CA, USA), and the peroxidase substrate DAB (Cat#: Vector SK4100, Vector Laboratories, Burlingame, CA, USA). Sections were then counterstained with hematoxylin/eosin.

### 4.8. RNA Extraction and Reverse Transcription-Quantitative Polymerase Chain Reaction (RT-QPCR)

Muscle samples (60 mg) were homogenized in a high-speed homogenizer (FastPrep-96™, MP Biomedicals) with 1 mL TRIzol reagent (Cat#: 15596018, Qiagen, Hilden, Germany) according to the manufacturer’s instructions, using iron 6 bead-beating tubes. The automated run was calibrated at 6.5 m s⁻¹ for 60 seconds. mRNA was isolated in 0.2 mL of chloroform, precipitated with 0.5 mL of isopropanol per mL TRIzol, and dissolved in 50 µL of RNase-free water overnight. Purity and cleavage were tested by optical density (Nanodrop, ThermoFisher, Waltham, MA, USA). Then, 8 µg of RNA was used for RT with the High-Capacity cDNA Reverse Transcription Kit (Cat#: 4368813, Applied Biosystems, Waltham, MA, USA) according to the manufacturer’s instructions (25 °C for 10 min, 37 °C for 2 h, 85 °C for 5 minutes). qPCR assays were performed in duplicate using the StepOnePlus Real-Time PCR System (Applied Biosystems, Waltham, MA, USA) with 7.5 µL of Master mix SensifFAST SYBR Hi-ROX kit (Cat#: BIO-92020, Bioline, France), 300 µM of forward and reverse primers (Table 2), and 2 µl of cDNA (20µl of final volume), and the following program: 20 s at 95 °C, then 30 seconds at 60 °C for 40 cycles. Results were normalized to invariant Cyclophilin A and rpS9 genes average, two genes that are always expressed in muscle. The cycle threshold value was used to compare the “Pre-DI” and “Post-DI” gene expression changes and results were calculated and expressed using the ΔΔCT formula.

### 4.9. Protein Isolation and Western Blotting

Muscle samples were homogenized in 10 volumes of lysis buffer [50 mM Tris pH 7.5, 150 mM NaCl, 1 mM EGTA, 1 mM EDTA, 10% Triton X-100, 5 mM Na3VO4, 1 M NaF, 1 M sodium dodecyl sulfate (10%), 50 mM ß-glycerophosphate and protease inhibitor cocktail 10 µL/mL (Cat#: P8380; Sigma-Aldrich, Saint-Louis, MI, USA) and centrifuged at 10,000 g at 4 °C for 10 minutes. Proteins were quantified with the Pierce BCA Protein Assay Kit (Cat#: 23225, ThermoScientific, Waltham, MA, USA). 50 µg of protein stock solution was mixed with 4X Laemmli buffer (Cat#: 1610747, Bio-Rad Laboratories, Inc., Hercules, CA, USA) and loaded on 4–20% SDS-polyacrylamide pre-cast stain-free gels (Cat#: 4568095; Bio-Rad, Hercules, CA, USA), followed by electrophoretic transfer onto nitrocellulose membranes (Cat#: 1704271, Bio-Rad; Trans-Blot Turbo Blotting System, Hercules, CA, USA). Membranes were blocked with 50 mM Tris-HCl (pH 7.5), 150 mM NaCl and 0.1% Tween 20 containing 5% skimmed milk or BSA for 1 h. They were then incubated with primary antibodies (Table 1) at 4 °C overnight, followed by the relevant peroxidase-conjugated secondary antibody (Table 1) at RT for 1 hour. After washing, membranes were incubated with ECL immunocruz (Cat#: sc-2048, Santa Cruz Biotechnologies, Inc., Santa-cruz, CA, USA) or with SuperSignals West Femto Maximum Sensitivity Substrate (Cat#: 34094, ThermoScientific, Waltham, MA, USA), and protein signals were visualized by enhanced chemiluminescence using the ChemiDoc Touch Imaging System. Images were then quantified with the Image Lab Touch Software (version 6.0). Stain-free total protein measurement was used as a loading control and used for normalization.

### 4.10. Statistical Analysis

All values are expressed as the mean ± SEM, and the significance level was set at *p* < 0.05. The Shapiro–Wilk test was used to confirm the normal distribution of the data. The paired Student’s *t*-test was used to compare Pre-DI and Post-DI data if normally distributed. The Wilcoxon signed-rank test was used for data that deviated from the normal distribution. Differences between Pre-DI and Post-DI and between groups were evaluated using two-way analysis of variance (ANOVA) matched pairs with Tukey HSD post-hoc or Friedman ANOVA for non-gaussian distributions. Statistical analyses and graphs were done with GraphPad Prism9.

## Figures and Tables

**Figure 1 ijms-23-05489-f001:**
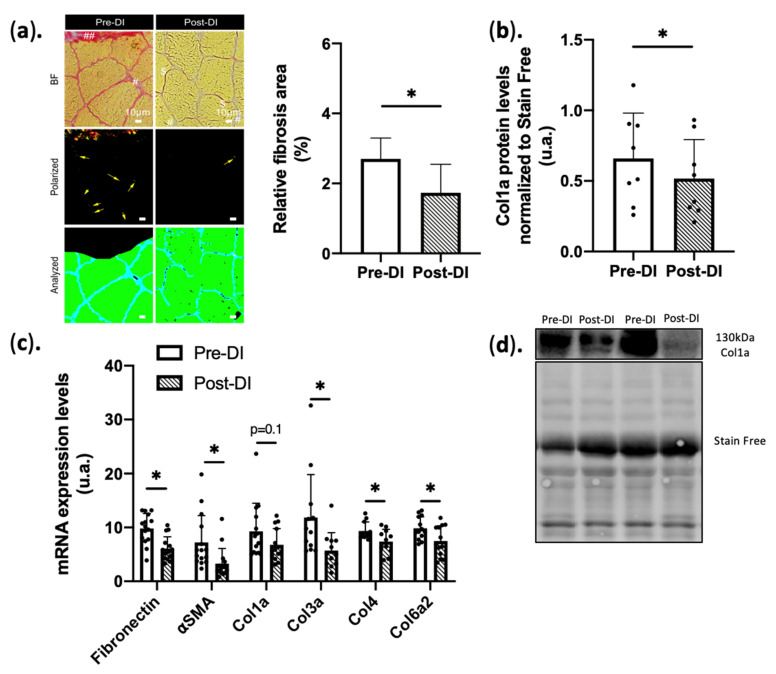
Extracellular matrix structure/components in VL biopsies before (Pre-DI) and after (Post-DI) 5 days of DI. (**a**) Sirius red staining in Pre-DI and Post-DI vastus lateralis biopsies; Sirius red-stained images analysis BF: Brightfield images; Polarized light images; Analyzed images Pre-DI vs. Post-DI; # shows automatically excluded area; ## shows manually excluded area; arrows shows fibrosis area; $ shows manually excluded area; scale bar measure 10 µm (n = 18/time point). Bar plot represents quantification of endomysium fibrosis area Pre-DI vs. Post-DI based on Sirius red analysis [32] (**b**) Collagen1a protein levels in Pre-DI and Post-DI muscle biopsies. n = 6/time point, data were normalized to the total protein staining using Stain-free technology ; (**c**) Gene expression levels of key fibrotic markers before and after 5 days of dry immersion n = 15/time point for Fibronectin, ⍺SMA, Col1a; n = 12/time point for Col3a/4/6a2; (d) Representative Western blot images and loading control; * *p* < 0.05. Paired *t*-test or Wilcoxon matched-pairs signed-rank test was performed according to normality.

**Figure 2 ijms-23-05489-f002:**
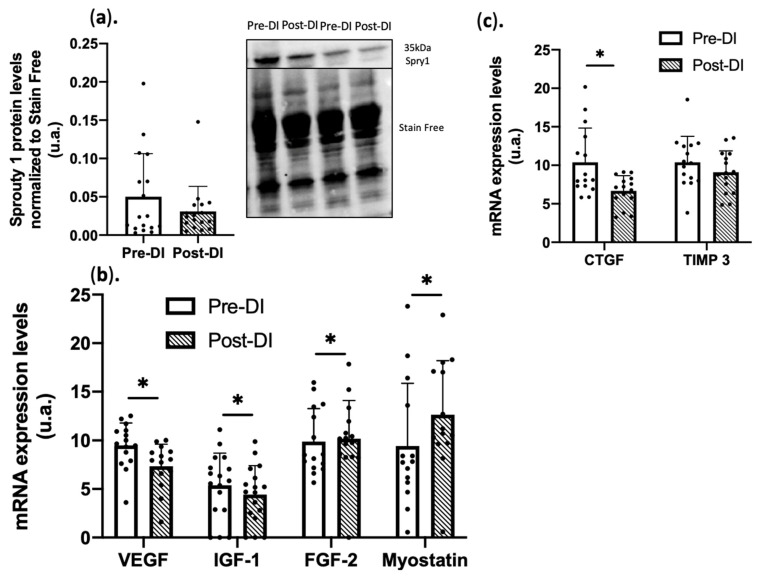
Key muscle microenvironment signaling components in VL biopsies before (Pre-DI) and after (Post-DI) 5 days of DI. (**a**) Sprouty 1 protein levels in Pre-DI and Post-DI muscle biopsies. n = 17/time point, data were normalized to the total protein staining using Stain-free technology; (**b**) Gene expression of ECM modulators in Pre-DI and Post-DI muscle biopsies. n = 14/time point for VEGF, Myostatin; n = 17/time point for IGF-1; n = 15/time point for FGF-2 (**c**) Gene expression of factors involved in the muscle microenvironment quality in Pre-DI and Post-DI muscle biopsies; n = 15/time point. * *p* <0.05, Paired *t*-test or Wilcoxon matched-pairs signed-rank test was performed according to normality.

**Figure 3 ijms-23-05489-f003:**
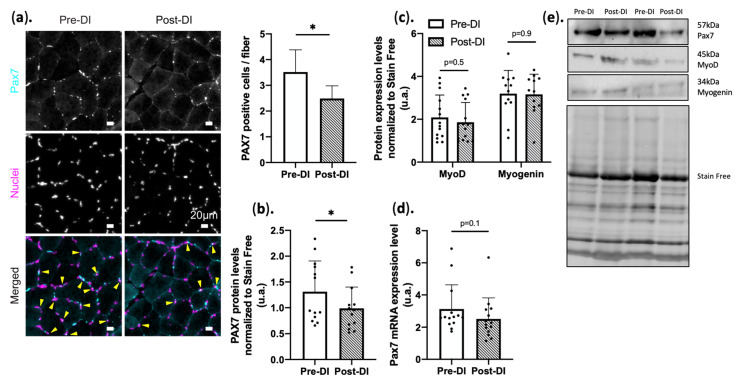
PAX7⁺ cell dynamics in VL biopsies before (Pre-DI) and after (Post-DI) 5 days of DI. (**a**) Representative images of Pre-DI and Post-DI vastus lateralis muscle sections stained with an anti-PAX7 antibody (cyan), Hoescht (Nuclei; Magenta), and Pax7/nuclei merged images; arrows correspond to colocalization of Pax7 and nuclei staining. Bar plot represents relative PAX7 signal after immunostaining paraffin-embedded transversal vastus lateralis sections from Pre-DI and Post-DI biopsies. (**b**) PAX7 protein levels in Pre-DI and Post-DI muscle biopsies, data were normalized to the total protein staining using Stain-free technology; n = 13/time point (**c**) MyoD and Myogenin protein levels in Pre-DI and Post-DI muscle biopsies, data were normalized to the total protein staining using Stain-free technology; n = 14/time point: MyoD; n = 12/time point: Myogenin (**d**) Gene expression of Pax7 in Pre-DI and Post-DI biopsies n = 14/time point; (e) Representative Western blot images and loading control ; * *p* < 0.05, Paired *t*-test or Wilcoxon matched-pairs signed-rank test was performed according to normality.

**Figure 4 ijms-23-05489-f004:**
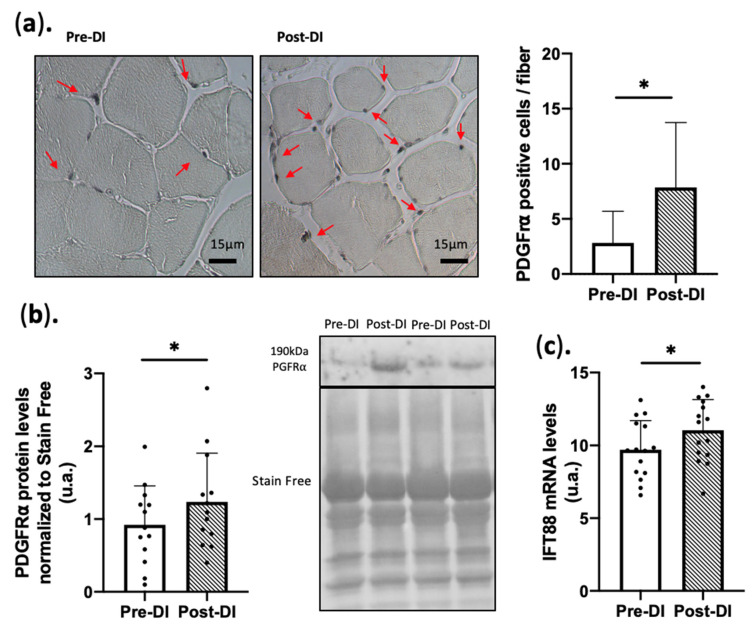
Expression of cell surface marker PDGFR⍺ in VL biopsies before (Pre-DI) and after (Post-DI) 5 days of DI. (**a**) Representative images of Pre-DI and Post-DI vastus lateralis muscle sections stained with an anti-PDGFR⍺ antibody; arrows show the staining localization. Bar plots represent relative PDGFR⍺ signal following staining of paraffin-embedded transversal vastus lateralis sections from Pre-DI and Post-DI muscle biopsies. (**b**) PDGFR⍺ protein level in Pre-DI and Post-DI muscle biopsies, data were normalized to the total protein staining using Stain-free technology. n = 13/time point (**c**) IFT88 gene expression levels in Pre-DI and Post-DI muscle biopsies. n = 15/time point; * *p* <0.05, Paired *t*-test or Wilcoxon matched-pairs signed-rank test was performed according to normality.

**Figure 5 ijms-23-05489-f005:**
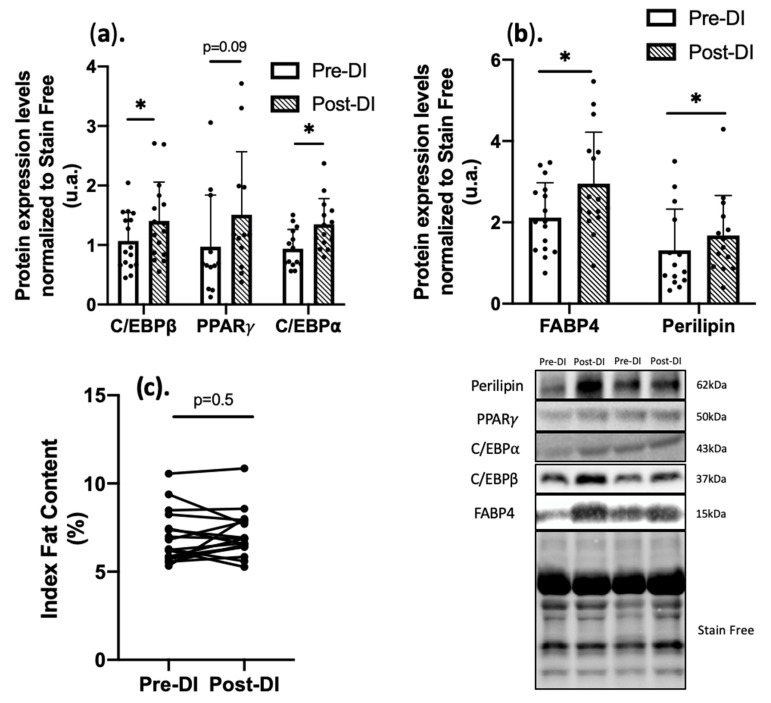
Expression of adipogenic differentiation markers in muscle from VL biopsies before (Pre-DI) and after (Post-DI) 5 days of DI. (**a**) Protein levels of key adipogenic markers in Pre-DI and Post-DI vastus lateralis biopsies. Data were normalized to the total protein staining using Stain-free technology (**b**) Protein levels of two mature adipocyte markers in Pre-DI and Post-DI vastus lateralis biopsies; data were normalized to the total protein staining using Stain-free technology. n = 16/time point for C/EBPβ, FABP4; n = 14/time point for C/EBP⍺, Perilipin; n = 12/time point for PPARg (**c**) Index Fat Content individual percent of Vastus lateralis measured by MRI before (Pre-DI) and after (Post-DI) 5 days of DI. * *p* < 0.05, Paired *t*-test or Wilcoxon matched-pairs signed-rank test was performed according to normality.

**Figure 6 ijms-23-05489-f006:**
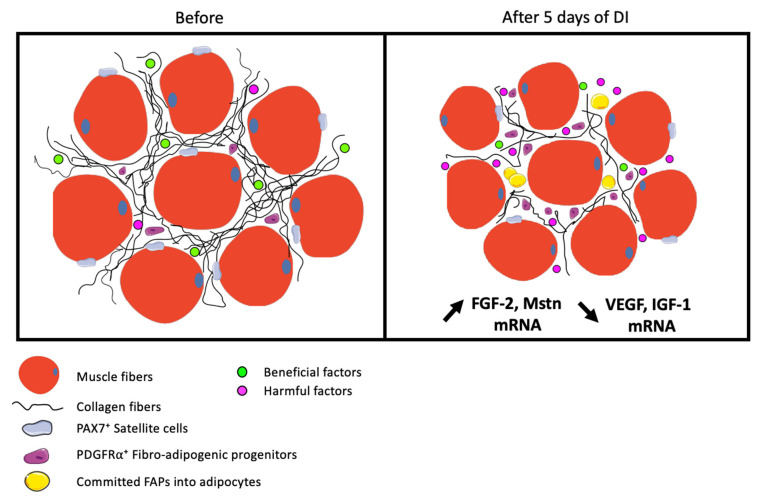
Muscle microenvironment degradation after 5 days of DI in healthy men. DI leads to a decrease in muscle fibers area as well as disorganization of ECM structure with reduced collagen and fibrosis content. After DI, ECM modulation, as well as atrophy, seems to drive a shift toward harmful signals resulting at least in FGF-2 and Myostatin signaling increases and VEGF and IGF-1 decreases. We hypothesized that these early changes dynamically stimulate muscle resident cells, impairing PAX7⁺ muscle progenitors (SCs) number and promoting PDGFRa⁺ Fibro-adipogenic progenitors (FAPs) proliferation and differentiation into mature adipocytes. Thus, ECM reorganization seems to be a critical regulator of resident cell fate and behavior.

**Table 1 ijms-23-05489-t001:** Western Blot/Immunostaining antibodies.

	Antibodies	References	Dilutions	Source
Primary	CEBP/α	#8178	1/500	Cell Signaling
	CEBP/β	sc-150	1/200	Santa Cruz
	Collagen1a	ab59435	1/200	Abcam
	FABP4	#3544	1/500	Cell Signaling
	Pax7	sc-81975	1/200	Santa Cruz
	PDGFRα	#3174	1/500	Cell Signaling
	Perilipin	#9349	1/500	Cell Signaling
	PPARγ	#2435	1/500	Cell Signaling
	Sprouty1	#13013	1/1000	Cell Signaling
	Laminin	L9393	1/200	Sigma-Aldrich
	MyHC 1	BA-D5	1/10	DSHB
	MyHC 2	M4276	1/200	Sigma-Aldrich
	MyHC 2a	SC-71	1/10	DSHB
Secondary	Alexa Fluor 568	A11031	1/800	Invitrogen
	Alexa Fluor 488	A11034	1/1500	Invitrogen
	Mouse	sc-2005	1/4000	Santa Cruz
	Rabbit	sc-2004	1/4000	Santa Cruz

**Table 2 ijms-23-05489-t002:** Real-time qPCR primers.

	Forward Primers	Reverse Primers	Amplicon Size (BP)
⍺SMA	ACGCTGAAGTATCCGA	CATTTTCTCCCGGTTGG	162
Collagen Ia	TCATCGTGGCTTCTCTGGTC	GACCGTTGAGTCCGTCTTTG	146
Collagen IIIa	CTTGATGTGCAGCTGGCATTCCTT	TCTCACAGCCTTGCGTGTTCGATA	267
Collagen IV	TTCCTGTACTGCAACCCTGGTGAT	ATATCCGATCCACAAACTCCGCCA	234
Collagen VIa2	AGCCTACGGAGAGTGCTACA	GTCCTGGGAATCCAATGGGG	173
CTGF	CTCCTGCAGGCTAGAGAAGC	GATGCACTTTTTGCCCTTCTT	94
Cyclophilin A	TTCCTCCTTTCACAGAATTATTCCA	CCGCCAGTGCCATTATGG	75
FGF-2	TGTGTCTATCAAAGGAGTGTGTGCTA	TCCGTAACACATTTAGAAGCCAGTA	84
fibronectin	CTGGCCGAAAATACATTGTAAA	CCACAGTCGGGTCAGGAG	114
IFT88	ATTGCCAATAGTTGTGGAGACTT	CTCGCTGTCTCACCAGGACT	88
IGF-1	GTGGAGACAGGGGCTTTTATT	CTCCAGCCTCCTTAGATCACA	122
Myostatin	TGCTGTAACCTTCCCAGGACCA	GCTCATCACAGTCAAGACCAAAATCC	113
P21	CCGAAGTCAGTTCCTTGTGG	CATGGGTTCTGACGGACAT	112
P53	GCTCAAGACTGGCGCTAAAA	GTCACCGTCGTGGAAAGC	128
rps9	CGGCCCGGGAGCTGTTGACG	CTGCTTGCGGACCCTAATGT	247
TIMP3	GCTGGAGGTCAACAAGTACCA	CACAGCCCCGTGTACATCT	72
VEGF	GCAGCTTGAGTTAAACGAACG	GGTTCCCGAAACCCTGAG	94

## Data Availability

The datasets used and analyzed during the current study are available from the corresponding author on reasonable request.

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
