# Peer review of "Severe Muscle Deconditioning Triggers Early Extracellular Matrix Remodeling and Resident Stem Cell Differentiation into Adipocytes in Healthy Men"

_ijms, 2022, doi:10.3390/ijms23105489_

Round 1

Reviewer 1 Report

Brief summary 

This publication consisted on dry immersion (DI) experiments of the human body mimicking  hypoactivity as the microgravity  in spaceflights or aging processes. 18 healthy volunteers were enrolled for several measurements including VL biopsies and MRI before and after 5 days DI.

The authors showed that this short DI (5 days) induced VL atrophy and strength loss. They hypothetised that DI induced alterations at molecular and cellular levels.

First, ECM remodeling decreased collagen 1a, fibronectin and ⍺SMA protein levels. The authors proposed a crosstalk regulation between FGF2 signaling and myostatin pathways.

Second, the authors showed discrepancies in cell populations with committant PAX7 positive cells population decrease, and PDGFRa positive cells increase after 5 days DI. They hypothesise the FAP progenitor population increase and early adipogenic differentiation.

General concept comments

To my point of view, the significance of the publication is highly improved in terms of both form and content. This revised version takes into account my major comments :  

  • The preliminary results that were already published have been switched to supplemental data ;
  • Concerning the figure format, all figures are organized in the same way, providing data from cellular (tissue imaging) to molecular phenotype, that improves result organisation and understanding ;
  • Concerning the experimental results, the barcharts show individual variations by displaying all data values instead of mean values (in the first manuscript) in each condition.

Specific comments

The experimental design clearly describes the composition of the control (9) and the cuff (9) groups. Since both groups are comparable, all data were pooled together, so n equals 18. More details are provided in supplementary data as previously described in the former paragraph entitled Dry immersion induces muscle atrophy and strength loss. These informations collected from both groups show the muscle physiology (fiber cross sectional area) and strength data (knee extension isometric torque).

The results are clearer now by starting with Sirius red staining, as in figure 1 for example, followed by imaging quantification and then molecular markers determinations (RNA and proteins).

It is much clearer to explain the imaging tool (automatically or manually delineation) in the methods part.

The addition of corresponding stain free blots confirms equal sample loading.

The figure subparts a) and b) are used twice (figure 1), but you could use them only once as a, b, c, d and e.

Author Response

We thank the reviewer 1 for his comments. We suppressed the duplicate figures ‘subpart

Reviewer 2 Report

The Authors demonstrate that after 5 days of microgravity by dry immersion, the ECM and the resident cell population derived from human muscle biopsies are significantly affected. These changes lead to a reduction of the Pax7+ positive cells, and toward a mature adipocyte phenotype of the FAPs cells.

The manuscript is clear and shows interesting data; the upload version contained the comments of the co-authors, which makes the reading a bit confused. Please next time make sure to upload the definite version. I would not recommend the use of the same letter for the figures.

Hereby some comments for the Authors:

The statistical analysis is not clear to me. If I understood correctly, paired tests was used to compare results derived from different samples (and maybe from different donors). I am not sure this is the best choice. Moreover it is not clear to me why the percentages of the comparison indicated in the text are without standard error.

Pax7+ population of Fig.3: I do not understand how the decrease of Pax7+ staining was quantified. If the aim is to demonstrate a reduction in the number, it should be better to measure it as numbers of Pax7+ cells (SCs) per fibers instead of signals per fibers. Moreover, why the level of the protein expression was reduced while the mRNA expression was the same (DI vs Ctrl) is not explained. I include the same observations for the quantification PDGFRs positive cells of Fig.4. I think is important to confirm the conclusion of the work.

Author Response

Reviewer 2:
Hereby some comments for the Authors:
The statistical analysis is not clear to me. If I understood correctly, paired tests was used to compare results derived from different samples (and maybe from different donors). I am not sure this is the best choice.
Moreover, it is not clear to me why the percentages of the comparison indicated in the text are without standard error.
Pax7+ population of Fig.3: I do not understand how the decrease of Pax7+ staining was quantified. If the aim is to demonstrate a reduction in the number, it should be better to measure it as numbers of Pax7+ cells (SCs) per fibers instead of signals per fibers. Moreover, why the level of the protein expression was reduced while the mRNA expression was the same (DI vs Ctrl) is not explained. I include the same observations for the quantification PDGFRs positive cells of Fig.4. I think is important to confirm the conclusion of the work.

A : We thank the reviewer 2 for his comments, About statistical analysis, data collection were from two biopsy of the same Vastus lateralis at two time point (before dry immersion = Pre-DI versus after = Post-DI). Therefore, the compared results in this study come from the same sample, this is why we performed a paired test.

The standard error for each percentage of Post-DI vs. Pre-DI comparisons was added in the new version Figures and legends corresponding to Pax7 and PDGFra quantification was redesigned to show the number of positive cells per fiber.
Despite a strong tendency the decrease of pax7 mRNA levels were not significant. We have no more explanation to reconcile these results.
According to Reviewer, the manuscript will be upload without “track resivion mode in this definite version. The duplicate figures subparts were suppressed in the new version.

Reviewer 3 Report

Overall, this is a really interesting study. It has used an excellent model of microgravty in dry immersion and is a high-quality study in the methods and analyses that have been employed. Unfortunately, the work is let down by perhaps “stretching” the data that have been collected. Just because a previous paper has shown no effect of your intervention (in this case, BFR) on your primary end-points, you cannot collapse across groups. This shows poor scientific practice and collectively with obvious flaws in the data such as gels that have not ran straight and poorly labelled figures, it does not give me confidence that these data or the findings would be reproducible. The ideas and analyses are interesting but should be studied in the placebo treatment arm only. Yes, the study would be less well powered but the results would be reproducible.

Major

If I’ve understood correctly the authors have collapsed across a cuff intervention and placebo. Whilst I understand the rationale, this is poor scientific practice. Just because the authors have not shown an effect in their previous publication, it doesn’t grant the right to collapse across an intervention; that first study could have been underpowered for the primary end points of interest. We therefore do not know if the effects presented in this paper could be driven by the cuff group for example through VEGF or HIF-1a or similar. It could be driven by a factor we do not even know to measure; hence the reason why we don't do it. The results from just the placebo treatment should be analysed, despite the shortcomings of the relatively low n number.

Minor

Results – need to state briefly why the two subjects were excluded more explicitly to reassure readers you are not manipulating the data pool

Figure 1 – labelling does not make sense - you have two a’s and two b’s

Figure 2 a blots – this gel needs to be re ran if it ran in this manner as it suggests the gel did not run evenly or an even current was not applied. These need to be repeated.

Figure 2 legend "n" needs to be lower case in all instances

Figure 3 b-c the gel has again not ran evenly – why? Does this suggest a consistent error with your gel tanks - why has this not been fixed after the first issue?

Figures all have multiple labels with a’s and b’s and c’s etc. Each panel within a figure needs a unique letter and the legend needs to accurately describe each panel

Methods – you state you used declaration of Helsinki so was informed consent written informed consent – if so, please amend

Page 27 author declaration – typo in “Early deconditioning” spelt “Eearly”

Author Response

.

Round 2

Reviewer 2 Report

The authors answered all my questions.
I noticed only that standard error values
of the precentuals inserted in the text are rather large.
I would recommend a further check.

This manuscript is a resubmission of an earlier submission. The following is a list of the peer review reports and author responses from that submission.

Round 1

Reviewer 1 Report

Guilhot et al. here report the influence of fast muscle deconditioning induced by dry immersion on muscle fiber microenvironment, i.e. fat infiltration and fibrosis. Authors overall show that short-term physical inactivity reproduced by a 5-day dry immersion model induces muscle atrophy, fat infiltration in the form of intermuscular adipose tissue and muscle fibrosis. This is overall an interesting work. However, there are a number of major issues as outlined below that preclude publication.

Major comments

  1. Authors state in introduction that “whether microgravity-induced hypoactivity can alter muscle microenvironment is not known” while in previous work, they already showed that microgravity and hypoactivity models in rodents are promoting the emergence of intermuscular adipose tissue (IMAT) and recruitment of fibro-adipogenic progenitors. For instance, authors previously published similar findings during a 3-day dry immersion model (PMID: 29248005). It is therefore unclear what is the novelty of this new study except that it is a 5-day DI model.

  1. What were authors expecting with the tigh cuffs countermeasure on their readouts? What was their hypothesis with respect to IMAT and muscle fibrosis? Is emergence of IMAT a consequence of muscle fiber atrophy or an independent event?

  1. Figure 2: author recently published data on muscle CSA from the same study (PMID: 34769492). They reported an overall 21.8% reduction of CSA while they here report the % change in each group. This is exactly the same data. The quality of images shown in (c) is suboptimal and better images should be provided.

  1. Authors should present MRI data of muscle CSA in a table or in a figure, not as “data not shown”.

  1. Figure 3 to 8. There is a large variability in reported data questioning the significance of the reported findings. Authors should clarify in each figure legend what kind of statistical test were performed to better appreciate if they are appropriate. Dot plot graphs should be shown for all figure to better appreciate the number of samples studied for each variable and data dispersion.

  1. Western blots are largely sub optimal for all target proteins measured and loading controls are not shown. Data must be reported after normalization to an appropriate loading control (housekeeping protein or stain free).

Author Response

Reviewer 1

Authors state in introduction that “whether microgravity-induced hypoactivity can alter muscle microenvironment is not known” while in previous work, they already showed that microgravity and hypoactivity models in rodents are promoting the emergence of intermuscular adipose tissue (IMAT) and recruitment of fibro-adipogenic progenitors. For instance, authors previously published similar findings during a 3-day dry immersion model (PMID: 29248005). It is therefore unclear what is the novelty of this new study except that it is a 5-day DI model.

RE: We agree the way we wrote the sentence could lead to such interpretation. By “microenvironment”, we meant essentially extracellular matrix (ECM) remodeling, knowing that previous work have establish that ECM composition could be related to changes in stem cells fate and or behavior (Garg et Boppart et al 2016 ; Stearn-Reider et al 2017).

Pointing out the previous work on dry immersion model (Pagano et al 2018) is the good example as we wanted to verify if FAPs behavior in this model could have been driven by ECM composition changes.

What were authors expecting with the tigh cuffs countermeasure on their readouts? What was their hypothesis with respect to IMAT and muscle fibrosis? Is emergence of IMAT a consequence of muscle fiber atrophy or an independent event?

RE: The present study has been set up by the French space agency (CNES) to assess on healthy male volunteers the effects of thigh cuffs to prevent the deconditioning induced by 5 days of dry immersion and in particular the fluid shift and its related ophthalmological disorders. Indeed, astronauts exposed to prolonged weightlessness experience hyperopic shifts and structural alteration in the eye (e.g., choroidal folds and optic disc edema). This condition was first referred to as Visual Impairment Intracranial Pressure (VIIP) syndrome and was recently redefined by NASA as Spaceflight-Associated Neuro-ocular Syndrome (SANS) (Balasubramanian et al., 2018). Changes in vision and eye structure are thought to be the result of prolonged exposure to space flight-induced headward fluid shifts and elevated intracranial pressure. Loss of the hydrostatic pressure gradient during spaceflight leads to this redistribution of body fluids toward the head. To prepare for future manned missions beyond the low Earth Orbit, the mechanisms underlying SANS syndrome have to be investigated and countermeasures designed to reverse or prevent SANS are required. Venoconstrictive thigh cuffs (VTCs) represent one possible countermeasure to mitigate a headward fluid shift. The space agencies are actively engaged in studying the initiation and progression of SANS syndrome through studies on the International Space Station (ISS) and on the ground, as in the present experiment using the dry immersion model.

Clearly, the objective of this study was to evaluate the neuro-ophthalmological impact of dry immersion and in particular the impact of intracranial pressure changes induced by the fluid shift as well as the preventive effect of thigh cuffs used to limit the head-ward fluid shift.

In any way the thigh cuff countermeasure was designed to prevent simulated microgravity induced musculoskeletal wasting. However, it was still pertinent to test whether or not prolonged wear of thigh cuff further worsened negative muscle adaptations, i.e. atrophy and force generation.

What we strive to show is that microgravity-induced hypoactivity and atrophy are accompanied by an activation of PDGFRa positive cell population and their engagement at least in part, into adipogenic lineage. It is difficult to conclude that muscle fiber atrophy per se is the cause of IMAT, however there is growing evidence to suggest a dialog between myocytes, satellite cells and FAPs keeping either FAPs into quiescent state or orienting them into fibrogenic or adipogenic lineage. That way, every change from one actor of this dialog, as myocyte atrophy could be, may break up the equilibrium and drive changes for the others actors (cell populations or ECM)

Figure 2: author recently published data on muscle CSA from the same study (PMID: 34769492). They reported an overall 21.8% reduction of CSA while they here report the % change in each group. This is exactly the same data. The quality of images shown in (c) is suboptimal and better images should be provided.

RE: The reviewer is right, we present the same CSA data as subjects of our study are the same than the previous publish paper. According to editor, data previously published by Fovet et al 2021 (Figure 1-2a-3a) were move in supplementary data (Figure S1 and Figure S2). Statement was added mentioning the reuse of the published design and data. However, It still important to us to recall the main impact of 5 days of DI upon muscle fiber surface area. These data are essential to understand the early perturbation of microenvironment and the perturbation that could occur between Myofibers, FAPs and satellite cells. We chose to present data on both groups to show there was no effect of thigh cuff countermeasure allowing us to pool muscle samples strengthening statistical power.

Authors should present MRI data of muscle CSA in a table or in a figure, not as “data not shown”.

RE: As suggested by the reviewer we add MRI CSA data in Figure S2

Figure 3 to 8. There is a large variability in reported data questioning the significance of the reported findings. Authors should clarify in each figure legend what kind of statistical test were performed to better appreciate if they are appropriate. Dot plot graphs should be shown for all figure to better appreciate the number of samples studied for each variable and data dispersion.

RE: We redid the figures using dot plot to appreciate data distribution. Except for figure S2, we put the statistical test used in the figure legend. As we performed two tailed paired t-Test, small differences between groups can give rise to statistical significance.

Western blots are largely sub optimal for all target proteins measured and loading controls are not shown. Data must be reported after normalization to an appropriate loading control (housekeeping protein or stain free).

RE: We have indeed specified that Western blot normalization had been performed using “stain free” images method. (see Methods section: western blotting, page 19) “Stain-free total protein measurement was used as loading control and normalization.”

We added the corresponding stain free images on each figure presenting Western blot results.

References:

- Stearns-Reider, K. M.; D’Amore, A.; Beezhold, K.; Rothrauff, B.; Cavalli, L.; Wagner, W. R.; Vorp, D. A.; Tsamis, A.; Shinde, S.; Zhang, C.; et al. Aging of the Skeletal Muscle Extracellular Matrix Drives a Stem Cell Fibrogenic Conversion. Aging Cell, 2017, 16 (3), 518–528. https://doi.org/10.1111/acel.12578.

- Fovet, T.; Guilhot, C.; Stevens, L.; Montel, V.; Delobel, P.; Roumanille, R.; Semporé, M.-Y.; Freyssenet, D.; Py, G.; Brioche, T.; et al. Early Deconditioning of Human Skeletal Muscles and the Effects of a Thigh Cuff Countermeasure. Int J Mol Sci, 2021, 22 (21), 12064. https://doi.org/10.3390/ijms222112064.

- Gigliotti, D.; Xu, M. C.; Davidson, M. J.; Macdonald, P. B.; Leiter, J. R. S.; Anderson, J. E. Fibrosis, Low Vascularity, and Fewer Slow Fibers after Rotator-Cuff Injury. Muscle Nerve, 2017, 55 (5), 715–726. https://doi.org/10.1002/mus.25388.

- Garg, K.; Boppart, M. D. Influence of Exercise and Aging on Extracellular Matrix Composition in the Skeletal Muscle Stem Cell Niche. J Appl Physiol (1985), 2016, 121 (5), 1053–1058. https://doi.org/10.1152/japplphysiol.00594.2016.

- Pagano, A. F.; Brioche, T.; Arc-Chagnaud, C.; Demangel, R.; Chopard, A.; Py, G. Short-Term Disuse Promotes Fatty Acid Infiltration into Skeletal Muscle. J Cachexia Sarcopenia Muscle, 2018, 9 (2), 335–347. https://doi.org/10.1002/jcsm.12259.

- Balasubramanian, S.; Tepelus, T.; Stenger, M. B.; Lee, S. M. C.; Laurie, S. S.; Liu, J. H. K.; Feiveson, A. H.; Sadda, S. R.; Huang, A. S.; Macias, B. R. Thigh Cuffs as a Countermeasure for Ocular Changes in Simulated Weightlessness. Ophthalmology, 2018, 125 (3), 459–460. https://doi.org/10.1016/j.ophtha.2017.10.023.

Reviewer 2 Report

Brief summary 

This publication consisted on dry immersion (DI) experiments of the human body mimicking  hypoactivity as the microgravity  in spaceflights or aging processes. 18 healthy volunteers were enrolled for several measurements including VL biopsies and MRI before and after 5 days DI.

The authors showed that this short DI (5 days) induced VL atrophy and strength loss. They hypothetised that DI induced alterations at molecular and cellular levels.

First, ECM remodeling decreased collagen 1a, fibronectin and ⍺SMA protein levels. The authors proposed a crosstalk regulation between FGF2 signaling and myostatin pathways.

Second, the authors showed discrepancies in cell populations with committant PAX7 positive cells population decrease, and PDGFRa positive cells increase after 5 days DI. They hypothesise the FAP progenitor population increase and early adipogenic differentiation.

General concept comments

This study aims to show molecular and cellular modifications occuring in the VL muscle after a short time in dry immersion. In these conditions, 2 parameters will make it difficult to prove the modifications : the short time inducing slight changes and the variability between the samples (18 muscles).

Specific comments

Dry immersion for 5 days leads to ECM remodeling

This first paragraph describes the 2 groups (control and cuffs). Differences are expected between the groups, but no effect is due to cuffs (atrophy or strength). This result is not discussed nor analysed. Has the lower-limb counterpressure (about 30-50 mmHg) been obtained ? Should the cuff be placed on both legs or only one and use the second one as control ? It seems that this preliminary result is out of the topic. If it is the case, it should be added as supplemental data.

In figure 2B, the myofiber cross-sectional area is significantly decreased in Post-DI. Is it possible to specify which types of fibers are impacted (type I, II or both) ?

In figure 2C : It is not clear wether the images have been stained or polarized.  Some white points are visible in Post-DI, are there artefacts ? Use a scale bar on each image, it will be homogeneous with your other images.

ECM signaling alterations

In figure 3A, the authors show that the Relative Sirius red stained surface area in Post-DI is significantly decreased. But the text says that « The total collagen surface area was not different in Pre-DI and post-DI muscle samples ». Is it correct ?

In figure 3B, BF Pre-DI image should be representative of healthy muscle. Is the endomysium fibrosis area often seen in VL healthy muscle ?  In BF Post-DI, it seems that myofiber cross-sectional area is increased according to the image, while it should be decreased (as shown in Fig2B). The quantification of staining that is detailled in the legend of Fig.3 should be reported and developped in the methods explaining which technical tool is used for excluding specific areas (automatically or manually). Add the length of the white scalebar in the legend.

In figure 3D : The WB samples are not clearly annotated. Are they 2 representative samples out of 6 (N=6/time point) ? Why 6 and not 18 samples as above ?

I don’t understand why the loading control is not shown. You should explain at least in the methods the reason why you used a Stain-free total protein measurement as loading control.

In Figure 4B and 4C, it is difficult to conclude that FGF2 signaling pathway plays any role (then summarized in Fig 8). There is indeed a weak difference (11%) at mRNA level but no data is shown at the protein level (FGF2, Myostatin) except sprouty 1 protein that is not significantly different.

PDGFRa expression is increased after 5 days of DI

In Figure 7B, the WB is overexposed and makes it difficult to conclude to perilipin overexpression. That would mean that adipogenic differenciation for 5 days generated mature adipocytes. But this was not seen on the images (Fig 6C). The possibility to generate myofibroblasts is not discussed. FAP progenitors may be committed to fibrogenic differenciation, that secretes collagen in a shorter time than for adipocyte generation. Myofibroblasts are also responsible for ECM remodeling (See reference below).

Arrighi N, Moratal C, Savary G, Fassy J, Nottet N, Pons N, Clément N, Fellah S, Larrue R, Magnone V, Lebrigand K, Pottier N, Dechesne C, Vassaux G, Dani C, Peraldi P, Mari B. The FibromiR miR-214-3p Is Upregulated in Duchenne Muscular Dystrophy and Promotes Differentiation of Human Fibro-Adipogenic Muscle Progenitors. Cells. 2021 Jul 20;10(7):1832. doi: 10.3390/cells10071832. PMID: 34360002; PMCID: PMC8303294.

Methods : Sodium orthovanadate should be written Na3VO4

Author Response

Reviewer 2

Dry immersion for 5 days leads to ECM remodeling

This first paragraph describes the 2 groups (control and cuffs). Differences are expected between the groups, but no effect is due to cuffs (atrophy or strength). This result is not discussed nor analysed. Has the lower-limb counterpressure (about 30-50 mmHg) been obtained ? Should the cuff be placed on both legs or only one and use the second one as control ? It seems that this preliminary result is out of the topic. If it is the case, it should be added as supplemental data.

As the thigh cuff countermeasure designed by the French space agency (CNES) was not dedicated to counteract microgravity induced muscle wasting, we were not surprised to find no difference between Control et Cuff groups. As said at the end of the paragraph “Dry immersion induces muscle atrophy and strength loss”: “As atrophy and muscle strength loss were not significantly different between groups, we arbitrarily chose to pool the data of all participants (n=18) to increase the statistical power”.

Indeed, as muscle atrophy and strength were similarly impacted by 5 days of DI, we no longer discriminate Control and Cuff subjects for the rest of biochemical assays performed.

The desired lower limb counterpressure was measured at each installation (every day).

As mentioned in Methods section, paragraph “Study design, dry immersion protocol, thigh cuffs” :

“Participants randomized in the Cuffs group wore thigh cuffs on both legs during the 5 days of DI: from 10 a.m. to 6 p.m. at day 1 of DI (DI1), and from 8 a.m. to 6 p.m. at DI2-DI5”, that specify thigh cuff were wore on both legs.

In figure 2B, the myofiber cross-sectional area is significantly decreased in Post-DI. Is it possible to specify which types of fibers are impacted (type I, II or both) ?

we decided not to insert data on what muscle fiber type was more affected in this model. When looking at muscle fiber type 1 or 2, our previous published study did not show any CSA decrease difference (Fovet et al. 2021).

We modified the sentence to state there was no difference between fiber types, referencing our previous study (Fovet et al. 2021).

In figure 2C : It is not clear wether the images have been stained or polarized.  Some white points are visible in Post-DI, are there artefacts ? Use a scale bar on each image, it will be homogeneous with your other images.

We improved the quality of images of the Figure (now) S2 changing the threshold view to fluorescent laminin staining images. We hope the figure will gain in clarity.

ECM signaling alterations

In figure 3A, the authors show that the Relative Sirius red stained surface area in Post-DI is significantly decreased. But the text says that « The total collagen surface area was not different in Pre-DI and post-DI muscle samples ». Is it correct ?

We apologize for the mistake: the Figure 3(b) (Now Figure 1(b)) quantified the relative fibrosis surface area and not total collagen surface area. (This is corrected now in the Figure). Thus we found a significant decrease in fibrosis area but unfortunately no change for the total collagen surface area although a tendency arose.

In figure 3B, BF Pre-DI image should be representative of healthy muscle. Is the endomysium fibrosis area often seen in VL healthy muscle ?  In BF Post-DI, it seems that myofiber cross-sectional area is increased according to the image, while it should be decreased (as shown in Fig2B). The quantification of staining that is detailled in the legend of Fig.3 should be reported and developped in the methods explaining which technical tool is used for excluding specific areas (automatically or manually). Add the length of the white scalebar in the legend.

As said above, basal level of reticulated collagen i.e. fibrosis exists in skeletal muscle (Gigliotti et al. 2016)

We adjusted the magnification of the Post-DI (BF) image to get the same than the Pre-DI one.

As suggested by the reviewer we precised in Method section that excluded areas have been manually drawn. For full methods description we refer to the published macro (ref 63).

The length of the white scalebar has been added to the legend.

In figure 3D : The WB samples are not clearly annotated. Are they 2 representative samples out of 6 (N=6/time point) ? Why 6 and not 18 samples as above ?

I don’t understand why the loading control is not shown. You should explain at least in the methods the reason why you used a Stain-free total protein measurement as loading control.

As suggested by the reviewer we changed for a more representative image.

Col1A Western blot has been performed on 6 subjects only because of limited biological materials. As significance appeared with only 6 subjects we are confident that extending subject number to 18 would have keep significance. Stain free loading control has been added.

In Figure 4B and 4C, it is difficult to conclude that FGF2 signaling pathway plays any role (then summarized in Fig 8). There is indeed a weak difference (11%) at mRNA level but no data is shown at the protein level (FGF2, Myostatin) except sprouty 1 protein that is not significantly different.

We fully agree with the reviewing this is why we stay careful with the scope of results and have changed “clearly showed” by “suggest” in the text (“Taken together, these data suggest a shift in the muscle microenvironment towards signals that may promote muscle atrophy, devascularization, and induction of FAPs differentiation.”)

PDGFRexpression is increased after 5 days of DI

In Figure 7B, the WB is overexposed and makes it difficult to conclude to perilipin overexpression. That would mean that adipogenic differenciation for 5 days generated mature adipocytes. But this was not seen on the images (Fig 6C). The possibility to generate myofibroblasts is not discussed. FAP progenitors may be committed to fibrogenic differenciation, that secretes collagen in a shorter time than for adipocyte generation. Myofibroblasts are also responsible for ECM remodeling (See reference below).

Arrighi N, Moratal C, Savary G, Fassy J, Nottet N, Pons N, Clément N, Fellah S, Larrue R, Magnone V, Lebrigand K, Pottier N, Dechesne C, Vassaux G, Dani C, Peraldi P, Mari B. The FibromiR miR-214-3p Is Upregulated in Duchenne Muscular Dystrophy and Promotes Differentiation of Human Fibro-Adipogenic Muscle Progenitors. Cells. 2021 Jul 20;10(7):1832. doi: 10.3390/cells10071832. PMID: 34360002; PMCID: PMC8303294.

The reviewer is right and results suggest that 5 days are sufficient to see increased content of mature adipocytes into skeletal muscle. The reviewer is also true saying images from Figure 6C do not show any adipocytes as they were chosen to depict FAPs population. The present results extend our previous study in which only 3 days DI were sufficient in seeing increased perilipin as well as FABP4 protein expression.

From in vitro studies we learn that perilipin expression is detectable from the first day of adipogenic induction medium. Thus one cannot exclude that some proliferating FAPs could have start to differentiate allowing us to detect significant perilipin expression using Western blotting.

We agree with the reviewer’s remark about the possibility of FAPs to generate myofibroblasts. However, from Figures 3(c) and 4(a) (Now Figures 1(c) and 2(a)) we learn that 5 days of DI decreased aSMA and CTGF mRNA levels. Although we did not confirm with protein expression, these results suggest that the main FAPs fate remain adipogenic but one cannot exclude that under prolonged conditions FAPs fate will move towards myofibroblast lineage, reconciliating thus increased collagen and fibrosis deposits retrieved in literature for longer period (Stearn-Reider et al 2017).

References:

- Stearns-Reider, K. M.; D’Amore, A.; Beezhold, K.; Rothrauff, B.; Cavalli, L.; Wagner, W. R.; Vorp, D. A.; Tsamis, A.; Shinde, S.; Zhang, C.; et al. Aging of the Skeletal Muscle Extracellular Matrix Drives a Stem Cell Fibrogenic Conversion. Aging Cell, 2017, 16 (3), 518–528. https://doi.org/10.1111/acel.12578.

- Fovet, T.; Guilhot, C.; Stevens, L.; Montel, V.; Delobel, P.; Roumanille, R.; Semporé, M.-Y.; Freyssenet, D.; Py, G.; Brioche, T.; et al. Early Deconditioning of Human Skeletal Muscles and the Effects of a Thigh Cuff Countermeasure. Int J Mol Sci, 2021, 22 (21), 12064. https://doi.org/10.3390/ijms222112064.

- Gigliotti, D.; Xu, M. C.; Davidson, M. J.; Macdonald, P. B.; Leiter, J. R. S.; Anderson, J. E. Fibrosis, Low Vascularity, and Fewer Slow Fibers after Rotator-Cuff Injury. Muscle Nerve, 2017, 55 (5), 715–726. https://doi.org/10.1002/mus.25388.

- Garg, K.; Boppart, M. D. Influence of Exercise and Aging on Extracellular Matrix Composition in the Skeletal Muscle Stem Cell Niche. J Appl Physiol (1985), 2016, 121 (5), 1053–1058. https://doi.org/10.1152/japplphysiol.00594.2016.

- Pagano, A. F.; Brioche, T.; Arc-Chagnaud, C.; Demangel, R.; Chopard, A.; Py, G. Short-Term Disuse Promotes Fatty Acid Infiltration into Skeletal Muscle. J Cachexia Sarcopenia Muscle, 2018, 9 (2), 335–347. https://doi.org/10.1002/jcsm.12259.

- Balasubramanian, S.; Tepelus, T.; Stenger, M. B.; Lee, S. M. C.; Laurie, S. S.; Liu, J. H. K.; Feiveson, A. H.; Sadda, S. R.; Huang, A. S.; Macias, B. R. Thigh Cuffs as a Countermeasure for Ocular Changes in Simulated Weightlessness. Ophthalmology, 2018, 125 (3), 459–460. https://doi.org/10.1016/j.ophtha.2017.10.023.